# How public reaction to disease information across scales and the impacts of vector control methods influence disease prevalence and control efficacy

**Jing Jiao** [1,2]*, **Gonzalo P. Suarez** [3], **Nina H. Fefferman** [1,4]

**1** National Institute for Mathematical and Biological Synthesis, The University of Tennessee, Knoxville, Tennessee, United States of America, **2** Department of Biological Science, Florida State University, Tallahassee, Florida, United States of America, **3** Department of Agriculture and Biological Engineering, University of Florida, Gainesville, Florida, United States of America, **4** Ecology & Evolutionary Biology, The University of Tennessee, Knoxville, Tennessee, United States of America

* jjiao3@utk.edu

**Data Availability Statement:** All results of this study come from software simulation. The R and Matlab code supporting the results is available

## Abstract

With the development of social media, the information about vector-borne disease incidence over broad spatial scales can cause demand for local vector control before local risk exists. Anticipatory intervention may still benefit local disease control efforts; however, infection risks are not the only focal concerns governing public demand for vector control. Concern for environmental contamination from pesticides and economic limitations on the frequency and magnitude of control measures also play key roles. Further, public concern may be focused more on ecological factors (i.e., controlling mosquito populations) or on epidemiological factors (i.e., controlling infection-carrying mosquitoes), which may lead to very different control outcomes. Here we introduced a generic Ross-MacDonald model, incorporating these factors under three spatial scales of disease information: local, regional, and global. We tailored and parameterized the model for Zika virus transmitted by *Aedes aegypti* mosquito. We found that sensitive reactivity caused by larger-scale incidence information could decrease average human infections per patch breeding capacity, however, the associated increase in total control effort plays a larger role, which leads to an overall decrease in control efficacy. The shift of focal concerns from epidemiological to ecological risk could relax the negative effect of the sensitive reactivity on control efficacy when mosquito breeding capacity populations are expected to be large. This work demonstrates that, depending on expected total mosquito breeding capacity population size, and weights of different focal concerns, large-scale disease information can reduce disease infections without lowering control efficacy. Our findings provide guidance for vector-control strategies by considering public reaction through social media.

from Zenodo: https://doi.org/10.5281/zenodo.
4728971.

**Funding:** NHF, JJ and GPS acknowledge financial
support from National Socio-Environmental
Synthesis Center (SESYNC) under funding received
from the National Science Foundation Division of
Biological Infrastructure (DBI)-1639145, and also
by The Division of Environmental Biology (DEB)-
1640951. The funders had no role in study design,
data collection and analysis, decision to publish, or
preparation of the manuscript.

## Author summary

With the development of modern technologies (e.g., social/mass media platforms), people
can access disease information across counties, states, or entire nations. This wider access
to information about (potentially remote) disease risks can motivate local citizenry to
demand rapid action to prevent/control exposure. In some cases, this demand may be
mismatched with actual risk (e.g., to enact control before disease is present locally, or dis-
proportionately to current prevalence). This paper first provides a systematic study about
the influences of larger-scale disease information on local-scale infection dynamics and
control effort and efficacy through a case study of Zika virus. We find that larger-scale
information often decreases local outbreak size. We find that the impact of information
on control efficacy depends on available vector breeding habitat. This study demonstrates
the importance of including likely public reaction to information about disease across spa-
tial and temporal scales into the design and implementation of disease control strategies.

## Introduction

For many vector-borne diseases, such as dengue, Zika virus, and West Nile Virus, vector-con-
trol is a common and effective way to reduce disease spreading in human populations [1–4].
For example, control of Zika virus disease frequently relies on controlling primary mosquito
vectors: *Aedes aegypti* [5]. Many mosquito-control practices or strategies are the responsibility
of local or regional governments (e.g., strategies incorporating host dynamics and economic
constraints; [6–8]. However, beyond governmental response, individual members of the public
can also voluntarily practice mosquito control by themselves [9,10].

Individual, voluntary mosquito controls largely depend on public perceptions of risk,
which in turn relies on the disease information available. With the development of modern
technologies, more and more individuals can easily get disease information through newspa-
pers, television, or other social media platforms [11–15]. These modern technologies can pro-
vide real-time information (whether accurate or not) across diverse spatial scales, ranging
from the area of residence (local) to a regional scale (local area and also neighboring areas), or
an even larger scale involving multiple areas where pathogen can possibly spread (formed as a
human metapopulation structure; hereafter referred to as global scale). This public perception
of disease information through social media can influence individual decisions and local pres-
sure to agencies about control actions. Often, members of the public demand control of mos-
quitoes in their own areas based on risk perception derived from disease information that
relies on larger spatial scales, even if this leads to inaccurate estimation of immediate local risk
[11,16]. For instance, they may start their individual mosquito control early if other places
have active Zika infections, even if their own areas do not report any Zika cases (i.e., before the
disease spreads to their areas). This could lead to anticipatory local mosquito control (hereafter
referred to as "early intervention";[17–19]. The larger numbers of human infections from
larger scales could also lead to tolerance of, or even demand for, the overuse of pesticides,
which increases the overall control strength in their local areas (hereafter referred to as
"strength reinforcement"). Because available breeding capacity in the environment can largely
affect mosquito population and potential disease prevalence [20,21], here we studied the
impact of public reaction to the above three scales of disease information under three levels of
environmental breeding capacity of mosquito.

Due to the economic and manpower costs associated with pesticide usage [22–24], another
associated public reaction would be the balance between larval control vs. adult control. Many

vectors such as mosquito have two life stages, which requires different control treatments. For example, mosquito larvae often live in standing water (e.g., ponds, birdbaths, rain puddles in standing garbage etc.), which involves water-based control [25–27]. Adult mosquitoes (the stage that interacts with humans by taking blood meals) are instead more sensitive to air-based control (e.g., pesticide spraying or fogging [28–30]). Although individuals and municipalities may have different functional limitations, they can be expected to consider trade-offs of mosquito control versus environmental contamination specific to the type of control involved (i.e., water vs air contamination). For example, with limited expense on pesticide purchase, if the public is more concerned about water contamination caused through larval control, they may be more willing to permit air contamination caused by air-based control, and vice versa to achieve mosquito control in keeping with their perception of relative risks to their own health and the health of their environment. This tradeoff can largely influence human infections from two aspects: vector population size and transmission rate. Larval control has been proved to be very effective to control mosquito population through regulating oviposition[6]. By lowering mosquito population size, larval control can reduce human infection by downsizing the disease reservoir (i.e., ecological control). Air-based control instead works directly on mosquito-human contact rate, so the control of the adult mosquito would directly decrease disease transmission rate (i.e., epidemiological control). Those two controls, combined with the scale of disease information, would affect mosquito life-history dynamics and human infections. We therefore also explored how the combinations of environmental concerns on both larvae and adult mosquito influence system equilibria given scales of disease information.

To understand all the above factors on disease control and prevalence, here we developed a simple vector-borne disease model under a human metapopulation framework, in which human can migrate among multiple patches, perceive disease information across scales (i.e., local, regional, and global scales), and then take control actions within each patch accordingly. For this study, we make the simplifying assumption that the vector—the mosquito—can only stay in their natal patch without migration (representing an inter-patch spatial scale beyond their natural range). In this system, we first studied the dynamics of average human infections, total control effort, and control efficacy per patch. We then included the tradeoff of environmental concerns relevant to larval versus adult controls to further explore the above results at equilibria.

## Methods

As an initial case, we assumed a human metapopulation (e.g., human urban system) with 9 patches (i.e., $N = 9$), which are occupied by both human and mosquito populations. The virus was initially introduced via one infected human in one focal patch, and then allowed to spread to other patches (this focal patch is randomly selected). We assume that local mosquitoes can only move within their natal patch, but humans can move between connected patches (see Fig 1). This assumption is consistent with the fact that mosquitoes *Aedes* seldom travel long distance [31–33] while human can easily travel among suburbs, cities, or regions. The human movement from patch $i$ to $j$ is defined as $m_{ij}$, so the total probability of moving from patch $i$ to all other patches would be $p = \sum_{j=1}^{N} m_{ij}$ (if there is no direct connection between the two patches, $m_{ij} = 0$). Initially, patch $j$ has $S_j^0$ susceptible human population, so the total human population in the system is $S^0 = \sum_{j=1}^{N} S_j^0$. For simplicity but without loss of generality, we assume all patches are randomly connected (here we used scale-free algorithm [34,35]: the patch structure was generated by initial patches as 5 and number of new added patches as 4).

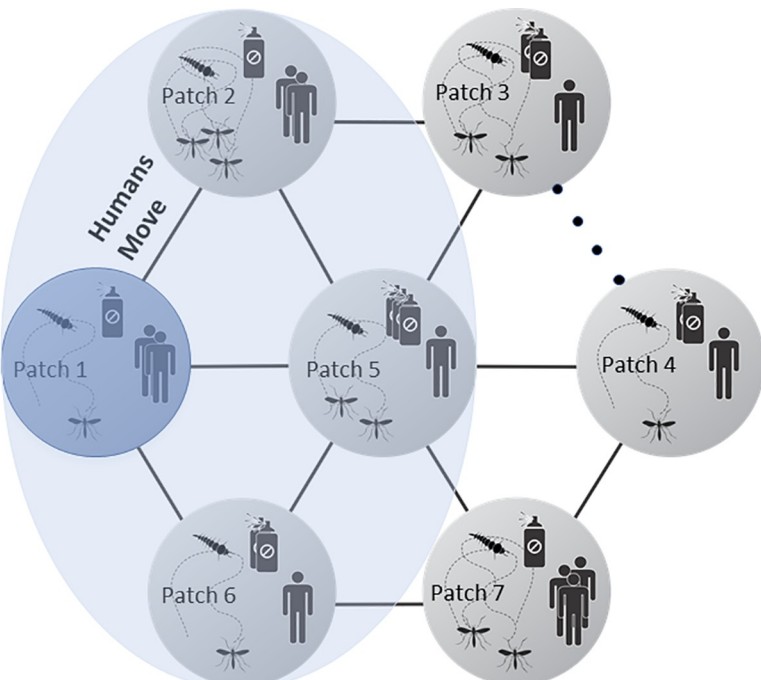

**Fig 1. The structure of human metapopulation with mosquito controls where control strength in each patch (e.g., focal patch 1) is based on human infections at a local scale (focal patch only in dark blue), regional scale (focal and neighbor patches: in blue) and global scale (all patches).** The dots between Patch 3 and 4 represent the potential patches in the system.

Within one patch at any time step, human can be in any of the three states: susceptible to infection ($S^H$), infected ($I^H$) or recovered ($R^H$). Some of the infected humans may develop serious symptoms (at a rate $\delta$), leading to the number of disease-attributable deaths ($D^H$). The total human population in this patch at that time would be $H = S^H + I^H + R^H$. We also considered the natural birth rate ($b^H$) and death rate ($\mu^H$) for human populations. The mosquito population has three states: larvae which cannot harbor the virus (i.e., we assume no vertical transmission) ($L^M$), uninfected adult mosquitoes ($S^M$) that are susceptible to pick up the virus through biting an infected human, and infected adult mosquitoes ($I^M$), which can transmit virus to susceptible human. The maturation rate from larvae to susceptible adult mosquitoes is $\nu$. The compound rate of biting and successful infection from susceptible mosquitoes and human are $\beta^M$ and $\beta^H$, respectively. The natural death rate of adult mosquitoes is $\mu^M$. The birth of larvae mosquitoes is limited by the patch breeding capacity (indicating the available breeding habitat), having the form $f\left(M, K_j\right) = M\left(1 - {}^M/_{K_j}\right)$ where $K_j$ is the breeding capacity of mosquito at patch $j$ and $M = S^M + I^M$, the total number of adult mosquitoes that lay eggs (Note, our model includes only on female mosquitos as they are both the individuals who take blood meals and generate eggs).

In each patch $j$, human and mosquitos' dynamics also drive the control actions for both larval ($C_j^L$) and adult mosquitoes ($C_j^M$) through usage of larvicides and adulticides. The public fear of disease, which drives control actions, increases with serious infected cases ($D^H$) and the number of infected people ($I^H$) but decreases with environmental concerns ($\epsilon^M$ and $\epsilon^L$: represent concerns for epidemiological risks and ecological risks, as controlled via focus on adult and larval mosquitoes, respectively). This concern provides a mechanism for negative feedback as control strength gets larger.

Here we propose a modified Ross-Macdonald equation [36,37] to capture the above dynamics in human, mosquitoes, and control actions in the system under Zika virus:

$$\frac{dS_j^H}{dt} = b^H H_j - \beta^H I_j^M S_j^H - \mu^H S_j^H + \sum_i m_{ij} S_i^H - S_j^H \sum_i m_{ji} \tag{1}$$

$$\frac{dI_j^H}{dt} = \beta^H I_j^M S_j^H - r I_j^H - \mu^H I_j^H - \omega^H I_j^H + \sum_i m_{ij} I_i^H - I_j^H \sum_i m_{ji} \tag{2}$$

$$\frac{dR_j^H}{dt} = r I_j^H - \mu^H R_j^H + \sum_i m_{ij} R_i^H - R_j^H \sum_i m_{ji} \tag{3}$$

$$\frac{dD_j^H}{dt} = \delta I_j^H \tag{4}$$

$$\frac{dL_j^M}{dt} = f\left(\eta(S_j^M + I_j^M), K_j\right) - v L_i^M - C_j^L L_j^M \tag{5}$$

$$\frac{dS_j^M}{dt} = v L_i^M - \beta^M I_j^H S_j^M - \mu^M S_j^M - C_j^M S_j^M \tag{6}$$

$$\frac{dI_j^M}{dt} = \beta^M I_j^H S_j^M - \mu^M I_j^M - C_j^M I_j^M \tag{7}$$

$$\frac{dC_j^M}{dt} = \alpha^M D_j^H + \gamma^M In_{scal}^H - \epsilon^M C_j^M \tag{8}$$

$$\frac{dC_j^L}{dt} = \left(\alpha^L D_j^H + \gamma^L In_{scal}^H - \epsilon^L C_j^L\right)\left(1 - \frac{C_j^M}{q + C_j^M + C_j^L}\right) \tag{9}$$

where Eqs 1–4 describe the dynamics of human population, Eqs 5–7 are for mosquito dynamics, while Eqs 8 and 9 indicate the dynamics of control actions on adult and larvae mosquitoes due to public reactions. We assumed a constant economic/manpower cost limit for total pesticide application (whether from larvicides or adulticides), and therefore assumed that the control change in larvae ($\frac{dC_j^L}{dt}$) is proportional to the percentage of the total control used for larval mosquitoes (i.e., $1 - \frac{C^M}{q} + (C^M + C^L)$ in Eq 9, where $q = 0.00001$ is used to prevent the denominator from getting to 0). At local scale, $In_{scal}^H = In_{local}^H = I_j^H$; at regional scale, $In_{scal}^H = In_{neigh}^H = I_j^H + \sum_k^n I_k^H$, in which patch $k$ indicates the adjacent patch to $j$ and $n$ is the number of all adjacent patches; for global scale, $In_{scal}^H = In_{global}^H = \sum_i^N I_i^H$, which is the sum of all infected humans across all patches $N$. In the absence of control (setting Eqs 8 and 9 = 0), the average human infections per patch is defined as $In_{Cont-}^H$. For simplicity, here we also assumed that each serious case produces 100 times the demand for control as that from one infected case (i.e., $\alpha^M = \alpha^L = 100\gamma$) and $\gamma^M = \gamma^L = \gamma$ [38]. The details of all the variables in the model are described in Table 1. The parameters and their values are in Table 2. The time $t$ is scaled to represent a single day. In the following, we use 300 timesteps (days) for all simulations, by which point the system would reach equilibrium (no change in each variable).

**Table 1. All variables and the corresponding initial values in patch j.**

| Variables | Description | Initial values/units |
|---|---|---|
| $S^H$ | Susceptible humans | 700; unit: no. |
| $I^H$ | Infected humans | 1 in a randomly chosen patch; 0 in other patches; unit: no. |
| $R^H$ | Recovered humans | 0; unit: no. |
| $D^H$ | Severe cases in humans | 0; unit: no. |
| $L^M$ | Mosquito larvae | 0; unit: no. |
| $S^M$ | Susceptible mosquitoes | 1000; unit: no. |
| $I^M$ | Infected mosquitoes | 0; unit: no. |
| $C^M$ | Control on mosquito adult | 0; unit: 1/time |
| $C^L$ | Control on mosquito larvae | 0; unit: 1/time |

Through this SIR model under human meta-population structure, we first analytically studied the general relationship between total control effort ($C_t^M + C_t^L$) and disease information across scales, including the potential effects from early intervention and strength reinforcement. Specifically, we simulated the dynamics of total control effort as well as the average human infections per patch under three information scales: local, regional, and global, and further calculated the corresponding control efficacy (i.e., the reduction of infection achieved per unit of control; $\frac{In_{Cont-}^H - In_{scal}^H}{C_t^M + C_t^L}$).

**Table 2. All parameters and their corresponding values in the model.** Some parameter values were chosen from the incidence and mortality in early Zika outbreaks in South America (see Reference).

| Parameters | Description | Value/units | Reference |
|---|---|---|---|
| $\beta^H$ | Transmission rate in humans | $1.5 \times 10^{-4}$; unit: 1/time | |
| $\beta^M$ | Transmission rate in mosquitoes | $3.0 \times 10^{-4}$; unit: 1/time | |
| $\mu^H$ | Natural mortality in humans | (8.6/1000)/365; unit: 1/time | Central, 2017 |
| $\mu^M$ | Natural mortality in mosquitoes | 1/13; unit: 1/time | Stone et al., 2017 |
| $b^H$ | Birth rate in humans | (9/1000)/365; unit: 1/time | Ellington et al., 2015 |
| $r$ | Recovery rate in humans | 0.037; unit: 1/time | Gao et al., 2016 |
| $\omega^H$ | Disease-induced mortality in humans | 0; unit: 1/time | |
| $\delta$ | Composite rate: the rate at which infection producing severe outcomes of the type that leads to increased public fear | 190/3, 474, 182; unit: 1/time | Ellington et al., 2016 |
| $v$ | Maturation rate | 1/7; unit: 1/time | Stone et al., 2017 |
| $\eta$ | Egg laying rate for mosquitoes | 10 | Stone et al., 2017 |
| p | Fraction of people traveling among patches | 0.01 | |
| q | a value to avoid mathematical insignificance | 0.00001 | |
| $\alpha^M$ | Control strength per severe case on adult mosquitoes | 100 times of $\gamma^M$; unit as 1/ (no. *time*time) | |
| $\alpha^L$ | Control strength per severe case on larval mosquitoes | 100 times of $\gamma^L$; unit as 1/ (no. *time*time) | |
| $\gamma^M$ | Control strength per infected case on adult mosquitoes | $e^{-\epsilon^M/80}$; unit as 1/ (no. *time*time) | |
| $\gamma^L$ | Control strength per infected case on larval mosquitoes | $e^{-\epsilon^L/80}$; unit as 1/ (no. *time*time) | |
| $\epsilon^M$ | Demotivation strength for adult mosquito control given per unit control | 100 or change as a variable with $\epsilon^M + \epsilon^L = 200$; unit: 1/time | |
| $\epsilon^L$ | Demotivation strength for larval mosquito control given per unit control | 100 or change as a variable with $\epsilon^M + \epsilon^L = 200$; unit: 1/time | |
| $K_j$ | Breeding capacity for mosquito larvae in patch j | $2000 \pm d$ where $d = unif\{1,10\}$; or change as other levels at 500, 800; unit: no. | |
| $m_{ij}$ | Human movement rate from patch $i$ to $j$ | 1/n (n is the number of all connected patches to focal patch $i$); unit: 1/time | |
| $In_{scal}^H$ | Infected human across scales | Change with $I_j^H$; unit: no. | |
| $In_{Cont-}^H$ | Average human infection per patch in the absence of control | Change with $I_j^H$; unit: no. | |

We also simulated the above results under three levels of mosquito breeding capacity (e.g., 500, 800 and 2000, representing small, medium, and large caps on mosquito populations) and the system equilibria under the tradeoff between environmental concerns on controlling mosquito larvae (or ecological control; $\epsilon^L$) or adult (or epidemiological control; $\epsilon^M$). To get stable results from the simulations, we excluded the patch where disease starts.

## Results

From Eqs 8 and 9, we can calculate the equilibrium control effort on both mosquito larvae and adult:

$$C_j^M* = \frac{1}{\epsilon^M}\left(\alpha^M D_j^H + \gamma^M In_{scal}^H\right) \tag{10}$$

$$C_j^L* = \frac{1}{\epsilon^L}\left(\alpha^L D_j^H + \gamma^L In_{scal}^H\right) \tag{11}$$

Thus, the total control effort at equilibrium is:

$$C_t^M* + C_t^L* = \frac{1}{\epsilon^M}\left(\alpha^M D_j^H + \gamma^M In_{scal}^H\right) + \frac{1}{\epsilon^L}\left(\alpha^L D_j^H + \gamma^L In_{scal}^H\right) \tag{12}$$

Because the disease information across scales depends on the infected human population at these scales (e.g., local information is determined by infected human in local patch, regional information depends on total infecteds in focal and neighboring patches; global information comes from the total infected ones at all patches), the small-scale disease information would be nested into larger-scale information at any time-step, i.e.,

$$0 \leq In_{local}^H \leq In_{neigh}^H \leq In_{global}^H, \tag{13}$$

and

$$\frac{d(C_t^M* + C_t^L*)}{dIn_{scal}^H} = \frac{\gamma^M}{\epsilon^M} + \frac{\gamma^L}{\epsilon^L} > 0, \tag{14}$$

so we would have total control effort $(C_t^M* + C_t^L*)$ positively correlated with the information of human infection across scales. The existence of control actions (Eqs 8 and 9), no matter under which scale of disease information $In_{scal}^H$, can largely reduce human infections.

### Scaled information with equal concerns in mosquitoes

When the public has a fixed and equal level of ecological as epidemiological concern (i.e., $\epsilon^M = \epsilon^L = \epsilon$, here we assume $\epsilon = 100$ for simulation), Eq 14 turns to:

$$\frac{d(C_t^M* + C_t^L*)}{dIn_{scal}^H} = \frac{\gamma^M + \gamma^L}{\epsilon} \tag{15}$$

Therefore, the change of total control effort has a fixed and constant relationship with the change in infected human cases across scales.

### Including early intervention

In the presence of early intervention, control action in a patch could start earlier than the time of the first local infected case (i.e., $C_t^M* + C_t^L*$ under the condition of $0 = In_{local}^H \leq In_{neigh}^H \leq In_{global}^H$). The verifies that control effort increases with the increase of information scale (see the increase

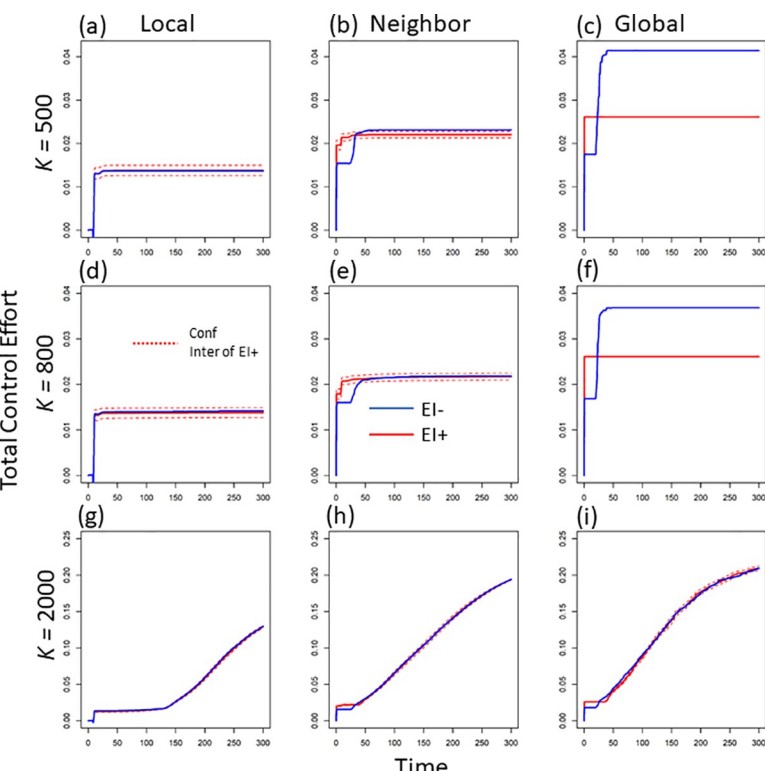

**Fig 2. The dynamics of total control effort (unit: 1/time) per patch under three levels of mosquito breeding capacity (e.g., K = 500, 800 and 2000) and three scales of disease information (i.e., local, regional, and global scale).** Here we exclude the patch where disease starts to get the average total control effort in one patch. The blue lines indicate the dynamic in the absence of early intervention, while the red lines are the dynamics in the presence of early intervention (the dashed red lines are the 95% confidence interval (CI)).

pattern of the red lines in Fig 2 from left to right). This pattern of the control effort exists across all three levels of mosquito breeding capacity (compare the red lines across rows in Fig 2).

No matter what scale of disease information, the introduction of control effort significantly decreases the average human infections per patch (compare the solid black line without control and the other lines with control in Fig 3). Under larger mosquito breeding capacity, stronger control effort from larger-scale information leads to larger reduction of average human infections in each patch (see the red-line trend across scales with $K = 2000$ in Fig 3G–3I). However, when mosquito breeding capacity is relatively low, there is no obvious relationship between information scale and average infected cases per patch (compare the red-line patterns across scales under $K = 500$ or $800$ in Fig 3).

The control efficacy (i.e., the decrease in human infection caused per unit of pesticide usage; $\frac{In^H_{Cont-} - In^H_{Scal}}{C^M_t + C^L_t}$), in general, decreases as the information scale increases (compare the red lines among the three columns at different mosquito carrying capacities in Fig 4A–4C, 4D–4F and 4G–4I). Although larger-scale information can largely reduce human infection under larger mosquito breeding capacity, the increase in total control effect still plays a larger role on shaping the control efficacy.

## Excluding early intervention

In the absence of early intervention (i.e., the control effort in each patch would not start until the disease arrives in that patch), control effort is defined to be 0 so long as infection remains

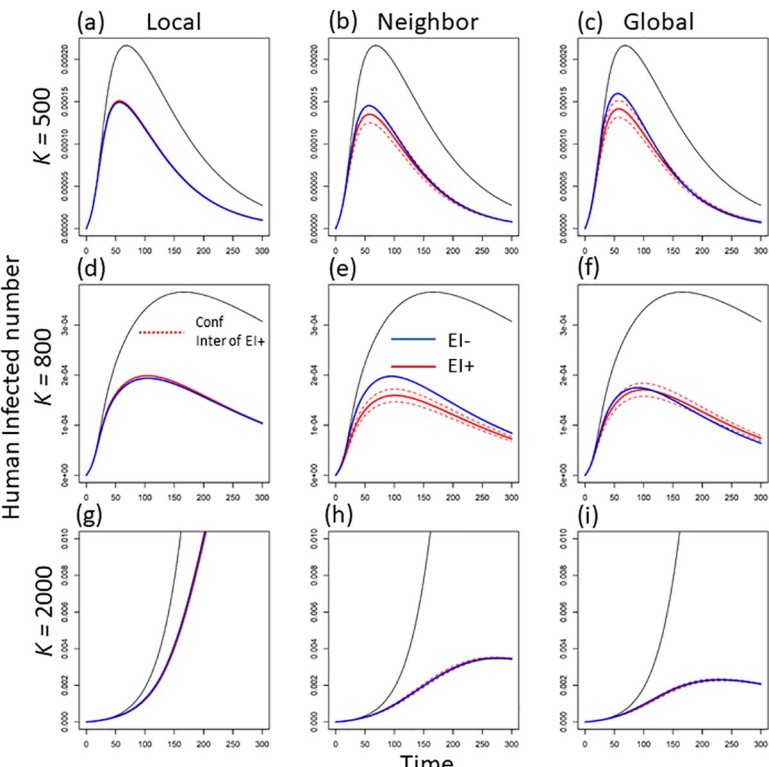

**Fig 3. The dynamics of human infections (unit: no.) per patch under three levels of mosquito breeding capacity (e.g., K = 500, 800 and 2000) and three scales of disease information (i.e., local, regional, and global scale).** Here we exclude the patch where disease starts to get the average infected human number in one patch. The blue lines indicate the dynamic in the absence of early intervention, while the red lines are the dynamics in the presence of early intervention (the dashed red lines are the 95% confidence interval (CI)).

at 0 (i.e., if $In_{local}^H = 0$, $In_{neigh}^H = In_{global}^H = 0$). This assumption largely increases the total control effort at larger-scale disease information compared to the scenario in the presence of early intervention (e.g., compare the red and blue lines at global scale under K = 500 or 800 in Fig 2C and 2F). However, the increased total control does not lead to the improved reduction in average human infections (compare the blue and red lines in Fig 3C and 3F). Compared to the control with early intervention, the average infected cases even show an overall increase in the absence of early intervention (compare the relative locations of the blue and red lines in Fig 3). Therefore, without early intervention, the increased control effort but lower infection reduction further leads to a lower control efficacy in general (compare the relative locations of the blue and red lines in Fig 4). The above inefficacy of control without early intervention is very strong when mosquitoes have lower breeding capacity (see the obvious decrease in efficacy under K = 500 and 800 in Fig 4). This means, under smaller mosquito breeding capacity, early intervention plays a large role in reducing the average human infections. This role of early intervention is specifically large when disease information is from a larger spatial scale (see the bigger difference of the red and blue lines in Fig 4C and 4F).

## Scaled information with trade-off concerns in mosquitoes

Because the above results showed that early intervention from larger-scale information can increase the control efficacy, in the following, we only analyze different concern combinations in the presence of early intervention.

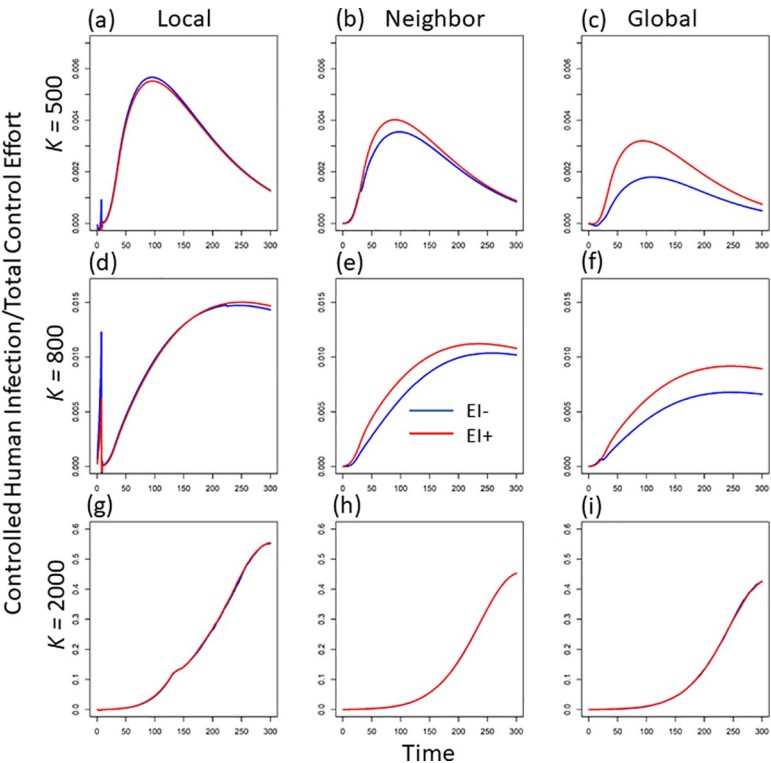

**Fig 4. The dynamics of control efficacy (i.e., the reduction in human infections per unit of total control effort; unit: no. * time) per patch under three levels of mosquito breeding capacity (e.g., K = 500, 800 and 2000) and three scales of disease information (i.e., local, regional, and global scale).** Here we exclude the patch where disease starts to get the average control efficacy in one patch. The blue lines indicate the dynamic in the absence of early intervention, while the red lines are the dynamics in the presence of early intervention.

When mosquito breeding capacity is small (e.g., $K = 500$), a relatively large environmental concern, limiting air-based control (i.e., epidemiological control), would significantly decrease infected cases in the human population (see the line trend in Fig 5A). Under an extremely small air-based environmental concern (i.e., larger usage of adulticide; epidemiological control), infected cases dramatically increase, especially under higher-scale disease information (compare the three lines around $\epsilon^M = 0$ in Fig 5A). This indicates that under a relatively small mosquito reservoir (i.e., smaller breeding capacity), a slight preference for larval control (i.e., usage of larvicide; or ecological control) would largely reduce disease prevalence. With the increase of mosquito breeding capacity (e.g., $K = 800$ and 2000), a unimodal pattern occurs: full environmental concern in either larvae ($\epsilon^L = 200$, $\epsilon^M = 0$) or adult mosquito ($\epsilon^L = 0$, $\epsilon^M = 200$) would lead to a dramatic decrease in human infections (see the hump curves in d and g in Fig 5). This demonstrates that under the larger mosquito population (i.e., larger breeding capacity), the mechanisms of infected reduction shift from shrinking vector population size (ecological control) to reducing the transmission rate (epidemiological control). This pattern holds true for all three scales of disease information, although the larger-scale information usually leads to a larger decrease in infected cases, which is consistent with the previous finding of fixed equal concerns (see Fig 3).

The total control effort ($C_t^M* + C_t^L*$) in general shows a unimodal pattern along the adult concern (see the larger values at both $\epsilon^M = 0$ and $\epsilon^M = 200$) across all information scales (see the U-shaped curves in Fig 5B, 5E and 5H). The larger information scale corresponds to bigger control efforts (compare the three lines in Fig 5B, 5E and 5H). This demonstrates that the

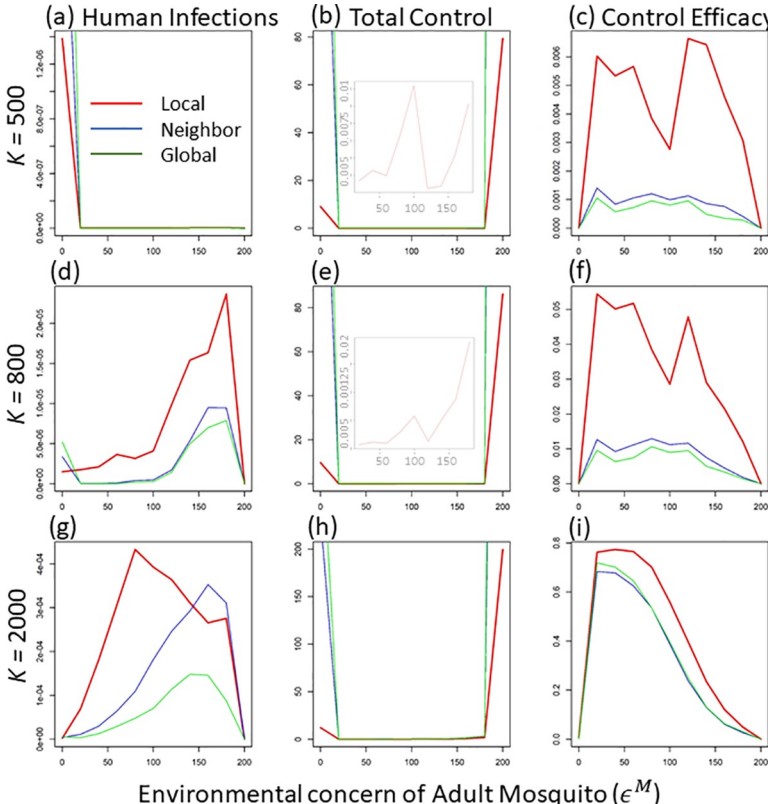

**Fig 5.** The average values of human infections (unit: no.), total control effort (unit: 1/time) and control efficacy (unit: no. * time) at equilibrium per patch (at time step = 300) along the environmental concern on adult mosquito ($\epsilon^M$) under three levels of mosquito breeding capacity (e.g., K = 500, 800 and 2000) and three scales of disease information (i.e., local, regional, and global scale, corresponding to the red, blue, and green lines). Here we exclude the patch where disease starts to get those average values. The embedded light plots in b and e show the enlarged graphs of the total control effort between 25 and 180 adult concern, in which all three lines (corresponding to three information scales) overlap.

higher infection reduction is usually coupled with larger control effort in general (as expected). This pattern is extremely strong when environmental concern is fully shifted to either water-based or air-based control.

The extreme large control effort at either full concern in larvae or adult mosquito further leads to a much lower control efficacy (see Fig 5C, 5F and 5I). In other words, if the public concern about pesticide usage only focuses on mosquito larvae or only on adult mosquitoes, the disease control would have very low efficacy (i.e., large pesticide usage without achieving concomitantly large disease reduction). When mosquito breeding capacity is relatively low (e.g., at K = 500 and 800), the control efficacy also shows a sharp drop near the middle level of adult concern (see $\epsilon^M = 100$ in Fig 5C and 5F), which corresponds the increase in total control effort at that concern level (see the embedded light graphs in Fig 5B and 5E). This is because, under lower breeding capacity, equal concerns (i.e., $\epsilon^L = \epsilon^M = 100$) could achieve a potentially higher mosquito population due to mosquito life-stage dynamics [4], thus, at equilibrium, larger control effort is needed to keep the vector size low (see the vector size at middle concern level when $\epsilon^M = 100$ in S2A and S2B Fig). In general, local-scale disease information achieves higher efficacy than both regional and global information (consistent with the fixed concern result; see the red lines in different columns of Fig 4). However, when mosquito breeding capacity is large (e.g., K = 2000), control efficacies across information scales do not show much difference,

especially when adult/ or epidemiological concern is relatively small (see the overlaps of the three lines in Fig 5I).

## Discussion

To design a more comprehensive and applicable control strategy for vector-borne disease, more and more studies have started to take human behavior into consideration [4,6,7,38–40]. In a modern media era, human behaviors are largely influenced by social media: e.g., the reports of infectious diseases via twitter or facebook, etc., which can affect the attitudes of the society towards disease control [14,41]. These impacts from social media have been gradually paid more and more attention as COVID-19 spreads [15,42]. Here we first studied how one important voluntary human reaction–public sensitive reactivity towards larger-scale disease information–can influence disease dynamics and control efficacy in a human metapopulation system.

   Compared to local-scale infection information, larger-scale disease information would trigger the public to impose stronger and earlier mosquito control in local areas. Under smaller mosquito breeding capacity (e.g., K = 500 or 800), mosquito population size (e.g., the similar maximum values across S1A–S1F Fig), mainly determines the human infections per patch (see the similar infected cases across Fig 3A–3F). This is due to large larval competition and mosquito population restriction under small breeding capacity [43,44]. When there is a larger vector breeding capacity, the environment does not meaningfully constrain the mosquito population (i.e., less larval competition under larger breeding capacity, thus, less restriction on the vector population size; [43,45]. In this case, the increase in information scale would decrease average infections via decreasing the number of adult mosquitoes as well as human-mosquito contact rate. The increased control effort with information scale mainly comes from strength reinforcement (i.e., the overall control strength in local patches; see the almost over-lapped blue and red lines in Fig 2). However, at the scale of global information, with relatively lower mosquito breeding capacity, early intervention can shrink the size of the mosquito reservoir (compare the red and blue lines in S1C and S1F Fig; see also Schwab *et al.* 2018), leading to a significant decrease in total control effort (compare the blue and red lines in Fig 2C and 2F).

   By impacting both the size of vector population and direct transmission rate, the concern preference between larval/ or ecological and adult/ or epidemiological control can further regulate human infections through mosquito life-stage dynamics. In general, smaller mosquito breeding capacity would affect infection through limiting the vector population size, while adult numbers and the transmission rate would play a major role in shaping human infections under a larger mosquito breeding capacity. Therefore, small concern about adult / or epidemiological control would decrease infections by lowering adult numbers and transmission rate (see the lower values at small x-axis values in Fig 5G), while small concern about larval / or ecological control (i.e., $\epsilon^L = 0$, $\epsilon^M = 200$) would decrease infections via shrinking the mosquito population size (see the lower size at higher x-axis values in S2 Fig). Similar to the fixed concern situation (e.g., $\epsilon^M + \epsilon^L = 200$), in general, larger-scale disease information (i.e., regional and global scale) and public sensitive reactivity leads to higher control effort but lower infections and control efficacy than local-scale information (compare the three scales in Fig 5B, 5E and 5H).

   Under lower mosquito breeding capacity, however, the average human infections are higher at lower levels of air-based concern (see the higher values of infections at low air-based concern in Fig 5A and 5D). This is because under small mosquito breeding capacity, mosquito larvae would have stronger density-dependent mortality (see Eq 5); thus, killing more adults

(i.e., less air-based concern) can release more space for mosquito larvae, which boosts mosquito growth (i.e., the compensation from density-dependent mortality; see [46–50]). With this compensated mosquito growth, average infections also increase (see the higher values at lower adult concern in Fig 5A and 5D). The large-scale disease information, which drives a higher control effort, would lead to stronger mosquito compensatory growth, as well as higher human infections (compare the three-scale lines at low mosquito concern in Fig 5A and 5D). Under the lower breeding capacity, equal concerns for both air- and water-based control would potentially lead to a higher vector population (through life-history dynamics; see Eqs 5 and 6); thus, at equilibrium, stronger control effort would be needed to reduce vector size (see the sharp increase in total control effort near the intermediate adult concern in Fig 5B and 5E). This large control effort at middle-level of adult concern further drives a sharp drop in control efficacy (see the drop in the middle of x-axis in Fig 5C and 5F). The density-dependent mortality, combined with control efforts to different life stages, can largely influence system dynamics and disease prevalence. Future disease control strategies targeting each life stage need to consider the above effects.

Different network structures as well as human migration (e.g., commute movement or directed migration to some "hub" areas; [51] can still influence this dynamics. In this paper, we assumed both an independent probability of connection among all patches and the random movement of a certain proportion of human among connected patches. Therefore, future studies can further explore those potential changes by relaxing those assumptions. We also assume that environmental concern does not change with control effort (see Eqs 5 and 6). However, in real systems, environmental concern might increase with the increase of control effort. For instance, the accumulated control effort (e.g., the usage of pesticide) may cause more serious environment degradation, leading to a larger negative feedback from the public. This would directly influence the dynamics of total control effort, as well as the evaluation of control efficacy. Further studies should be done to explore the sensitivity of information scale to this non-linear concern-control correlation. Here we also assume all local patches perform similar control activities given the same disease information. However, public reaction in different areas may show heterogeneous variation. For example, some hub areas may be more likely take actions based on global-scale disease information, while some rural areas might tend to take actions based on local-scale disease information.

In addition, the control efficacy in this study can also be easily modified to fit different applications. For example, the impact of the local economy could be included in the control effort to evaluate the infection reduction per unit of pesticide expense: e.g., higher cost of pesticide would lower the efficacy given certain economic expense. In that case, for areas with access to cheaper pesticide, control efficacy would tend to be larger, so the increase of control effort in certain range may not be the main consideration for implementing disease control. In areas with more expensive pesticides, we would expect relatively lower efficacy. Furthermore, local government may incorporate public sensitive reactivity into their control strategies [41,52], which may lead to a different format for control efficacy. Insecticide resistance [53] could also largely influence the control efficacy, which may also affects larval and adult mosquito differently. Further studies can be done to consider these types of factors.

In summary, our study demonstrates that, although the public reaction due to the disease information from larger-scale disease information may decrease the overall control efficacy, the amount of this decrease depends on mosquito breeding capacity and the combinations of environmental concern about air-versus water-based pesticide use. Our findings can be broadly applied into real system, and further guide local government to anticipate and leverage the potential negative influences from public sensitive reactivity and further achieve integrated vector management [54–56]. For example, at the locations where mosquitoes have large

suitable habitats (e.g., ponds, unremoved trash piles, small vessels for standing water, etc.; see [21,26]; where mosquito has larger breeding capacity), control agencies could guide local members of the public to slightly shift their control effort towards air-based control of adult mosquitoes (e.g., through social media; [57–59]. For the areas where mosquitoes have only limited suitable habitat (i.e., lower mosquito breeding capacity), encouraging local members of the public to adopt early intervention would largely decrease the disease reservoir and achieve a higher control efficacy (see the red lines in Fig 4A–4F).

## Supporting information

**S1 Fig. The dynamics of mosquito population size (larvae + adult) within 300 time steps under the influences of information scale (i.e., local, regional, and global) and mosquito breeding capacity.** The blue lines indicate the dynamics in the absence of early intervention while the red lines show the mosquito size in the presence of early intervention.
(TIFF)

**S2 Fig.** The mosquito population size (larvae + adult) at equilibrium (at time 300) along with the change of environmental concerns on adult mosquito ($\epsilon^M$) under the influences of information scale (i.e., local, regional, and global) and mosquito breeding capacity. The red lines indicate local scale, the blue lines are for region-scale information, and the green lines are for global scale.
(TIFF)

**S1 Appendix.**
(DOCX)

## Author Contributions

**Conceptualization:** Jing Jiao.

**Methodology:** Jing Jiao, Gonzalo P. Suarez.

**Project administration:** Nina H. Fefferman.

**Resources:** Nina H. Fefferman.

**Software:** Jing Jiao, Gonzalo P. Suarez, Nina H. Fefferman.

**Supervision:** Nina H. Fefferman.

**Validation:** Jing Jiao.

**Writing – original draft:** Jing Jiao.

**Writing – review & editing:** Jing Jiao, Gonzalo P. Suarez, Nina H. Fefferman.

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
