## [Decision Letter · Decision Letter 0]

29 Mar 2021

Dear Dr. JIAO,

Thank you very much for submitting your manuscript "How public reaction to disease information across scales and the impacts of vector control methods influence disease prevalence and control efficacy" for consideration at PLOS Computational Biology. As with all papers reviewed by the journal, your manuscript was reviewed by members of the editorial board and by several independent reviewers. The reviewers appreciated the attention to an important topic. Based on the reviews, we are likely to accept this manuscript for publication, providing that you modify the manuscript according to the review recommendations.

Sincerely,

Alex Perkins

Associate Editor

PLOS Computational Biology

Virginia Pitzer

Deputy Editor-in-Chief

PLOS Computational Biology

[LINK]

Reviewer's Responses to Questions

**Comments to the Authors:**

Reviewer #1: Good subject material.

I think that the TITLE should have "model" added to it - either at the end or beginning.

Please include in the discussion section commentary on insecticide resistance with regard to the impact of this model - any speculation (possibly before the concluding paragraph)?

Please describe on the Figures (1,2,and 3) what the "Time" interval unit is.

Please describe on Figure 4 what the Y-axis units are and what is a "concern level of adult mosquitoes". I was unaware how 1 was different from 200.

Please make the following minor corrections:

Line 51: remove "s" from "...largely depends..." (largely depend)

Line 54: add "the" between ",or Internet" ("or the Internet")

Line 69: "available breeding environment breeding capacity" - should both those breedings be in that sentence or can one be cut out?

Line 75: mosquitoes do not have 2 life stages, they have 4.

Line 76: remove "the" from "live in the standing water".

Line 81: reduce "contamination based that are specific to the type" to "contamination specific to the type"

Line 86: remove "concerns" in "This tradeoff concerns can..."

Line 87-88: Cite the statement: "Larval control has proven to be very effective to control mosquito population" and give the stipulation.

Line 91: insert "the" in "...control of adult mosquito..." to read "...control of the adult mosquito..."

Line 337: insert ")" after "(46)" to close the parenthetical statement started in line 336.

Multiple concerns:

Please elaborate on the statement in line 350-351: more expensive pesticides would expect lower efficacy... ? I would not necessarily agree with that statement - is there an assumption not discussed regarding cost of control measures?

Reviewer #2: Review is uploaded as an attachment.

Reviewer #3: This paper raises an interesting question, and models fairly simply yet convincingly the impact of public attitudes towards a raising threat by a vector-borne disease such as Zika.

A few remarks nonetheless:

- even from the abstract on, the authors focus on "social media" and its impact. Yet, the model does not seem to address specifically this problem. There is no explicit modeling, for instance, of a "viral" transmission of information on a social media platform. Here, the focus is mostly on public attitudes ("reactions" if you will) towards a new epidemics in their area or not that far from them. But these reactions could be driven by anything else (traditional media, rumors etc.), and the model would still be correct.

- the "overreaction" term seems unsuitable for the situation described. Individuals are indeed correct in assuming that their situation is connected to the situation in an area close to them. Talking about "overreaction" would then imply that there is a proper level of reaction, i.e. an adequate value for "risk aversion".

- the conclusion regarding impact on public policy seems overly optimistic, as the the realism of this model seems still precarious. A few assumptions are still very strong, some mechanisms are ignored (no explicit way to determine whether "overreaction" will occur in a certain epidemic or not ; no feedback between actions taken by the government / the population and the attitudes towards the epidemic ; no sens of the politisation of a crisis - whatever the authorities recommend is perfect/disastrous depending on your political opinion etc.)

This article is however a good first step towards a better understanding of these situations.

Reviewer #4: The overall messaging of the study is interesting. Calculations and conclusions drawn from model agree with underlying theme of the work.

This thesis is nicely argued, I have some minor points:

The omega (line 145) parameter in the main equations 1-9 is not explained in the text. Perhaps that’s a typo and is meant to be delta? If not then I do not see any transfer rates from infectious compartment (I_H)to seriously infected compartment (D^H). Some typos in Eqns 7-9 probably. If not for the typos then the notation for control parameters and control change in larvae/adult mosquitoes is confusing.

Notation of total control effort line 170 with equations 8-9 could be better, specially given hat small 't' is used for time, Again some inconsistency with notation between parameters in the table and in equations--makes for a difficult reading.

Equations in line 193 could be written in separate lines perhaps, right now equation numbering has crept into the main text. Make the notation of I_scale (from lines 159 to 153) consistent with the notation used for local, regional and global scale used at line 193, Eqs. 13 and 14.

Figure 4(c) and the text explaining it is currently not consistent. There is a dip in 4c at local scale, could that be explained somehow? Also the legends, labels, and captions need to be improved for all figures. It is very hard to understand them in the current format.

The results subsections could use better titles, my suggestions would be: Early intervention and delayed intervention. I leave to the authors to come up with better titles.

**Have all data underlying the figures and results presented in the manuscript been provided?**

Reviewer #1: Yes

Reviewer #2: Yes

Reviewer #3: Yes

Reviewer #4: Yes

PLOS authors have the option to publish the peer review history of their article (what does this mean?). If published, this will include your full peer review and any attached files.

Reviewer #1: No

Reviewer #2: No

Reviewer #3: No

Reviewer #4: No

Figure Files:

Data Requirements:

Reproducibility:

References:

---

## [Decision Letter · Decision Letter 1]

28 May 2021

Dear Dr. JIAO,

We are pleased to inform you that your manuscript 'How public reaction to disease information across scales and the impacts of vector control methods influence disease prevalence and control efficacy' has been provisionally accepted for publication in PLOS Computational Biology.

Best regards,

Alex Perkins

Associate Editor

PLOS Computational Biology

Virginia Pitzer

Deputy Editor-in-Chief

PLOS Computational Biology

Reviewer's Responses to Questions

**Comments to the Authors:**

Reviewer #2: The authors have adequately addressed the points I raised and I have no further comments.

Reviewer #3: Thank you for the answers and the few changes made.

Reviewer #4: I am satisfied with the changes made

**Have the authors made all data and (if applicable) computational code underlying the findings in their manuscript fully available?**

Reviewer #2: Yes

Reviewer #3: Yes

Reviewer #4: Yes

PLOS authors have the option to publish the peer review history of their article (what does this mean?). If published, this will include your full peer review and any attached files.

Reviewer #2: No

Reviewer #3: No

Reviewer #4: No

---

## [Editor Report · Acceptance letter]

22 Jun 2021

PCOMPBIOL-D-21-00102R1 

How public reaction to disease information across scales and the impacts of vector control methods influence disease prevalence and control efficacy

Dear Dr JIAO,

I am pleased to inform you that your manuscript has been formally accepted for publication in PLOS Computational Biology. Your manuscript is now with our production department and you will be notified of the publication date in due course.

With kind regards,

Katalin Szabo
